# Distinct mutational signatures characterize concurrent loss of polymerase proofreading and mismatch repair

N.J. Haradhvala[1,2], J. Kim[2], Y.E. Maruvka[2], P. Polak [1,2,3], D. Rosebrock[2], D. Livitz [2], J.M. Hess[2], I. Leshchiner [2], A. Kamburov[1,2,3], K.W. Mouw[3,4], M.S. Lawrence[1,2,3] & G. Getz [1,2,3]

Fidelity of DNA replication is maintained using polymerase proofreading and the mismatch repair pathway. Tumors with loss of function of either mechanism have elevated mutation rates with characteristic mutational signatures. Here we report that tumors with concurrent loss of both polymerase proofreading and mismatch repair function have mutational patterns that are not a simple sum of the signatures of the individual alterations, but correspond to distinct, previously unexplained signatures: COSMIC database signatures 14 and 20. We then demonstrate that in all five cases in which the chronological order of events could be determined, polymerase epsilon proofreading alterations precede the defect in mismatch repair. Overall, we illustrate that multiple distinct mutational signatures can result from different combinations of a smaller number of mutational processes (of either damage or repair), which can influence the interpretation and discovery of mutational signatures.

[1] Department of Pathology and Cancer Center, Massachusetts General Hospital, 55 Fruit Street, Boston, MA 02114, USA. [2] Broad Institute of Harvard and MIT, 415 Main Street, Cambridge, MA 02142, USA. [3] Harvard Medical School, 25 Shattuck Street, Boston, MA 02115, USA. [4] Department of Radiation Oncology, Brigham and Women's Hospital, Dana-Farber Cancer Institute, 450 Brookline Ave, HIM 350, Boston, MA 02215, USA. These authors contributed equally: Haradhvala NJ, Kim J, Maruvka YE. These authors jointly supervised this work: Mouw KW, Lawrence MS, Getz G. Correspondence and requests for materials should be addressed to G.G. (email: gadgetz@broadinstitute.org)

Cells have evolved multiple mechanisms to ensure that DNA replication occurs in a timely and accurate manner. However, DNA replication fidelity is frequently compromised in cancer, leading to accumulation of somatic mutations throughout the evolution of the tumor. These mutations serve as a footprint of the tumor's underlying DNA damage and repair landscape[1,2]. Specific mutational signatures have been identified in tumors with mutations in the exonuclease (proofreading) domain of polymerase epsilon (POLE), as well as tumors with mutations or epigenetic silencing of genes in the mismatch repair pathway (MMR)[3,4]. Until recently, it was thought that simultaneous loss of both POLE or polymerase delta (POLD1) proofreading and MMR function could not be tolerated by cells due to excessive accumulation of mutations[5–7]. However, a recent analysis of tumors from children with biallelic germline MMR deficiency (bMMRD) revealed a subset of tumors with remarkably high mutation burdens (>250 mutations/Mb) that also had a somatic mutation in POLE or POLD1[8]. In a follow-up study, concurrent to our own, the mutational spectra observed in a large number of adult tumors were clustered, revealing that a subset of endometrial and colorectal cancers have spectra similar to the pediatric bMMRD POLE-mutated cases[9]. In addition, a study of more than 500 endometrial tumors revealed that nearly half (12/30) of the tumors with a POLE exonuclease domain mutation also displayed microsatellite instability (MSI)[10], the characteristic patterns of insertions and deletions associated with loss of MMR function. Here, we investigate the mutational properties of tumors with POLE/POLD1 proofreading loss and MSI. We identify mutational signatures associated with concurrent POLE or POLD1 exonuclease mutations (POLE-exo* or POLD1-exo*) and MMR loss that are distinct from the signatures associated with either event in isolation.

## Results

**Proofreading and MMR deficiency in endometrial cancer.** To characterize the relationship among POLE/POLD1 proofreading, MMR function, and mutational signatures, we first sought to identify tumors with simultaneous loss of both pathways. Endometrial tumors are known to possess relatively high rates of both events, so we began our search in The Cancer Genome Atlas

(TCGA) cohort of 531 endometrial (UCEC) samples[3]. Experimental MSI status (MSI-high [MSI-H], MSI-low [MSI-L], or microsatellite stable (MSS)) was available for 410 tumors[11], and we identified 15 samples classified as MSI-H that also harbored a somatic point mutation in the POLE exonuclease domain (codons 268–471[12]), as well as eight MSI-H samples with no POLE proofreading mutation but instead possessing a POLD1 exonuclease domain mutation (codons 304–517[12], Supplementary Data 1 and 2). The majority of these 23 tumors displayed large numbers of microsatellite insertions and deletions (MS indels) indicative of MMR loss[13] as well as a high point mutation burden consistent with loss of polymerase proofreading (Fig. 1a). Many of these cases (7/15 POLE-mutated MSI-H tumors and 5/8 POLD1-mutated MSI-H tumors) were found to have epigenetic silencing of MLH1, the most common mechanism that causes loss of MMR in tumors, suggesting that the high MS indel rates observed in these tumors were indeed due to MMR deficiency. The majority (7/8) of POLE and all (3/3) POLD1-mutated tumors classified as MSI-H but lacking MLH1 silencing had either a truncating/frame shift mutation or deletion of one of the four major MMR pathway genes (MLH1, MSH2, MSH6, or PMS2), although due to the high mutation rate of POLE, many tumors may have such mutations in MMR genes by chance.

Tumors with either of the two most common POLE exonuclease domain mutations (P286R or V411L) were rarely MSI-H (2/30 cases), whereas the majority of tumors with any other POLE-exo mutation were MSI-H (13/20) (Fisher's exact test $p = 1.6 \times 10^{-5}$; Fig. 1b). Interestingly, POLE-exo* tumors with a V411L mutation were often classified as MSI-L (7/12 cases) despite having MS indel burdens that resembled MSI-H samples with a wild-type (WT) POLE/POLD1 exonuclease domain. In contrast, most MSI-L tumors with either a P286R POLE mutation or with WT exonuclease domain did not have high numbers of MS indels (Fig. 1b). Therefore, whereas P286R and WT POLE samples were likely classified as MSI-L due to chance mutation of a locus assayed in the Bethesda test, the V411L POLE samples might actually represent bona fide MMR-deficient (i.e., MSI-H) cases that were misclassified as MSI-L by the Bethesda assay, which was developed and validated in colon tumor cohorts (in which concurrent POLE/POLD1 proofreading loss and MSI has not been observed).

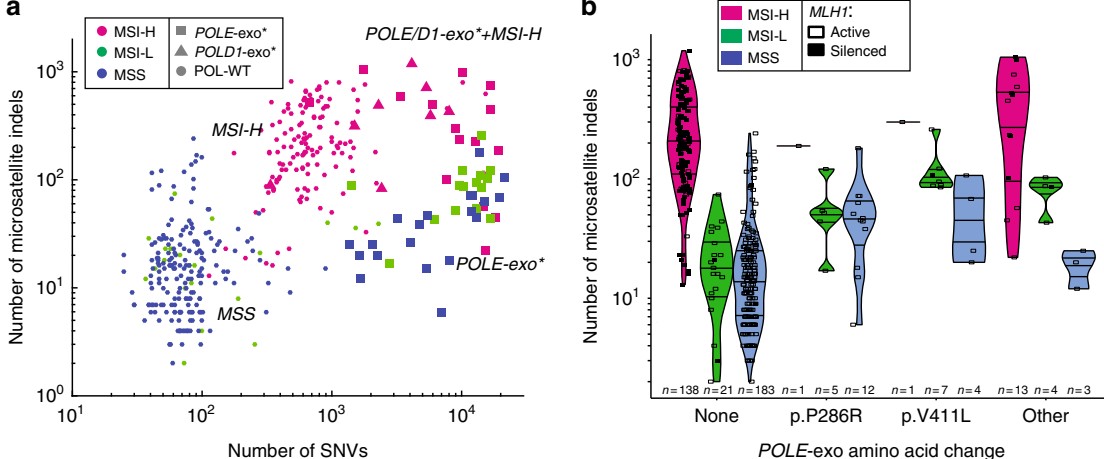

**Fig. 1** The landscape of single-nucleotide variants (SNVs) and microsatellite indels in endometrial cancer. **a** Number of SNVs and microsatellite indels for tumors in the TCGA endometrial cohort (UCEC). Colors represent the TCGA microsatellite classification determined using the Bethesda protocol,[11] and shapes represent the mutation status of POLE and POLD1 exonuclease domains. All samples with available Bethesda classification and MS indel calls are shown (n = 401). **b** Number of microsatellite indels for endometrial tumors with different POLE-exo* mutations. Color represents MSI status, and tumors with MLH1 epigenetic silencing are displayed as filled boxes. P286R and V411L are the two most common POLE exonuclease domain mutations in the cohort. All samples with Bethesda classification, MS indel calls, and MLH1 silencing calls available are shown (n = 392)

We next aimed to determine whether *POLE/POLD1*- and MSI-mediated mutagenesis occurred simultaneously in the same cancer cells, or whether the events were separated spatially (i.e., occurring in distinct subclones) or temporally (i.e., defective in non-overlapping time intervals). First, we noted that *POLE/POLD1* exonuclease domain mutations were frequently clonal (13/15 *POLE* and 8/10 *POLD1* mutations had an estimated cancer cell fraction [CCF] > 0.9) and that non-clonal *POLE/POLD1* mutations appeared to be non-functional passenger events (Supplementary Figure 1). This suggests that the *POLE/POLD1* proofreading defect was present in all cancer cells, and therefore, present in the same cells as the MMR deficiency.

**Mutational signatures of replicative repair deficiency**. We next reasoned that if the MMR and polymerase proofreading deficiencies were not present simultaneously, then the joint mutational signature should appear as an additive combination of the signatures corresponding to the individual defects. Alternatively, if the processes were active simultaneously—that is, if the cells had loss of both polymerase proofreading and mismatch repair at

the same time—they could potentially yield a distinct mutational signature reflecting the effect of combined loss. To test this hypothesis, we applied our SignatureAnalyzer tool[14,15] that discovers mutational signatures by applying a Bayesian variant of non-negative matrix factorization[14,16] to mutation data from a collection of tumors (Methods section).

SignatureAnalyzer identified 12 mutational signatures in the set of 531 endometrial samples (Supplementary Figures 2, 3; Supplementary Data 1, 3; Methods section). To explore these 12 signatures, we correlated their activities (i.e., the estimated number of mutations contributed by each signature to the total mutational burden of the tumor) to the MSI status and *POLE/POLD1* exonuclease domain mutation status of each tumor in the cohort. Eight of the 12 signatures were associated with deficiencies in MMR and/or *POLE/POLD1* proofreading (4/12 showed no such enrichment; Supplementary Figure 4). Cases with a *POLE* exonuclease mutation had enrichment of one of three signatures (Fig. 2a, b), designated as signatures E1–E3. Signatures E1 and E2 closely resemble the signature previously associated with somatic *POLE* exonuclease mutations (COSMIC

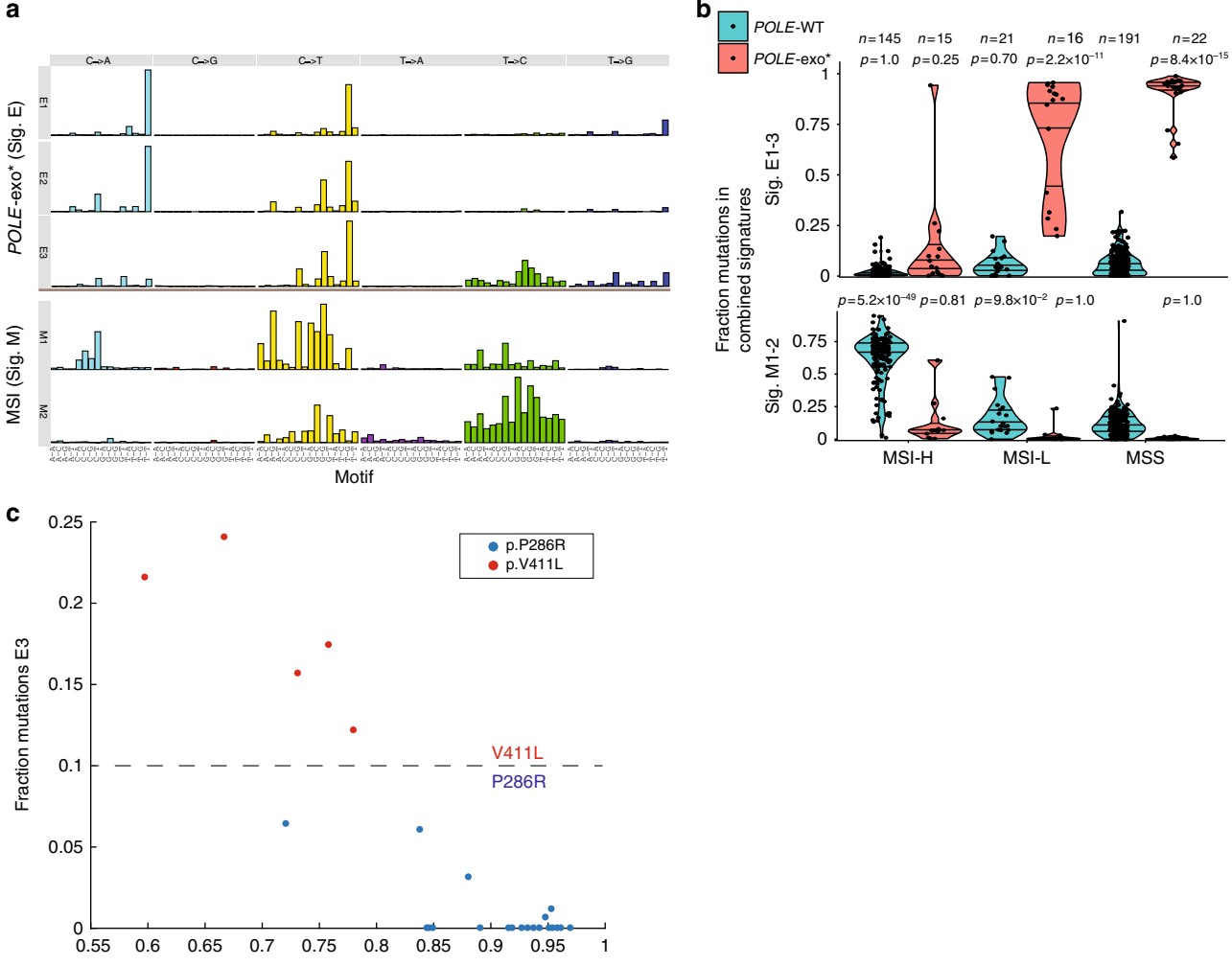

**Fig. 2** Identification of mutational signatures associated with individual loss of *POLE* proofreading or MMR. **a** SignatureAnalyzer yielded five mutational signatures associated with loss of polymerase proofreading (E1, E2, and E3) or mismatch repair (M1 and M2). For each mutational signature, the number of mutations is plotted for the 96 trinucleotide mutational contexts. **b** The fraction of mutations assigned to each signature across patients, segregated by *POLE*-exo mutations and Bethesda microsatellite classification. The distribution of each group is compared to that of polymerase wild-type MSS samples using a one-tailed rank-sum test. Horizontal lines within each group represent quartile values. **c** Relative contribution of E1 and E3 signatures to 5 V411L and 20 P286R samples with spectra dominated (>75% contributions) by signatures E1–3. Tumors with the P286R hotspot mutation have primarily contributions from E1, whereas tumors with a V411L hotspot mutation have additional contributions from E3

signature 10; Supplementary Figure 5)[1], whereas signature E3 was associated with the *POLE* V411L hotspot mutation (Fig. 2c). Similarly, we identified two signatures, designated as signatures M1 and M2, characteristic of MSI-H tumors (Fig. 2a, b). These signatures correspond to COSMIC database signatures 6 and 26, respectively, and have previously been noted in MSI samples (Supplementary Figure 5)[1].

**Signatures of concurrent proofreading and MMR deficiency.** We next turned our attention to the 15 samples with paired *POLE* exonuclease domain mutation and mismatch repair deficiency (*POLE*-MSI). Despite the presence of both defects in these samples, we observed minimal contributions from signatures E1–3 and M1–2 (Fig. 2b, signatures E1–3 contributed <20% mutations in 12/15 samples, and signatures M1–2 <20% in 12/15 samples). Instead, two distinct signatures—signatures C1 and C2—were highly enriched in these *POLE*-MSI tumors (Fig. 3a, b). Signatures C1 and C2 shared certain features with the individual MSI (M1–M2) and *POLE* (E1–E3) signatures but also had distinct motifs such as a unique G[C>A]T peak (Fig. 3c and Supplementary Figure 6). These unique motif features may represent mutations that can be repaired by either *POLE* proofreading or MMR and thus only accumulate in tumors lacking both repair functions (Fig. 3d).

Comparing the C1 and C2 signatures to the COSMIC signature database, we found that, when summed using coefficients fit to maximize the cosine similarity, signatures C1 and C2 closely resembled a signature of unknown etiology, COSMIC signature 14 (Fig. 3e), more than expected by chance (Supplementary Figure 7a, b and Supplementary Data 4). Although the activity of signatures C1 and C2 were correlated in our analysis ($R^2 = 0.26$), some tumors exhibited high activity of signature C1 relative to signature C2 (Supplementary Figure 8), demonstrating that indeed two distinct signatures are active in these samples.

Finally, in addition to identifying signatures of concurrent *POLE*-exo* and MMR deficiency, we found an additional signature (designated as signature D1) that was highly active (>1000 mutations and >30% of mutation burden) in 4/10 MSI-H tumors that also had a mutation in the exonuclease domain of *POLD1* (Fig. 4a, b). This signature closely resembled COSMIC signature 20 (Fig. 4a and Supplementary Figure 5), which had been associated with MMR deficiency but had not been previously linked to *POLD1* proofreading deficiency. In the four tumors with a *POLD1* mutation and high levels of signature D1, the mutation was located within the exonuclease domain (Fig. 4c, d), including residues D316 and E318 within the exonuclease active site. Mutations at the homologous position of D316 in yeast had been shown to impair exonuclease activity[17,18], and a D316N mutation was identified in an ultramutated gastic tumor[19]. An S478N mutation was also associated with high signature D1 activity, and the same mutation in the germline has been associated with increased risk of early-onset colorectal cancer[20]. In one case with high signature D1 activity, there was no exonuclease mutation, but the tumor did have an L606M mutation in the *POLD1* polymerase domain (Fig. 4c, d), which had been shown in vitro to increase polymerase delta error rate[21] and had also been observed in hypermutated glioblastomas with germline biallelic mismatch repair deficiency[8,9]. In the remaining six *POLD1*-exo* samples with MSI-H, two also had a *POLE*-exo mutation and strong contributions from signatures C1 and C2, making it difficult to conclude whether the mutations do not impact *POLD1* exonuclease function or are overwhelmed by the

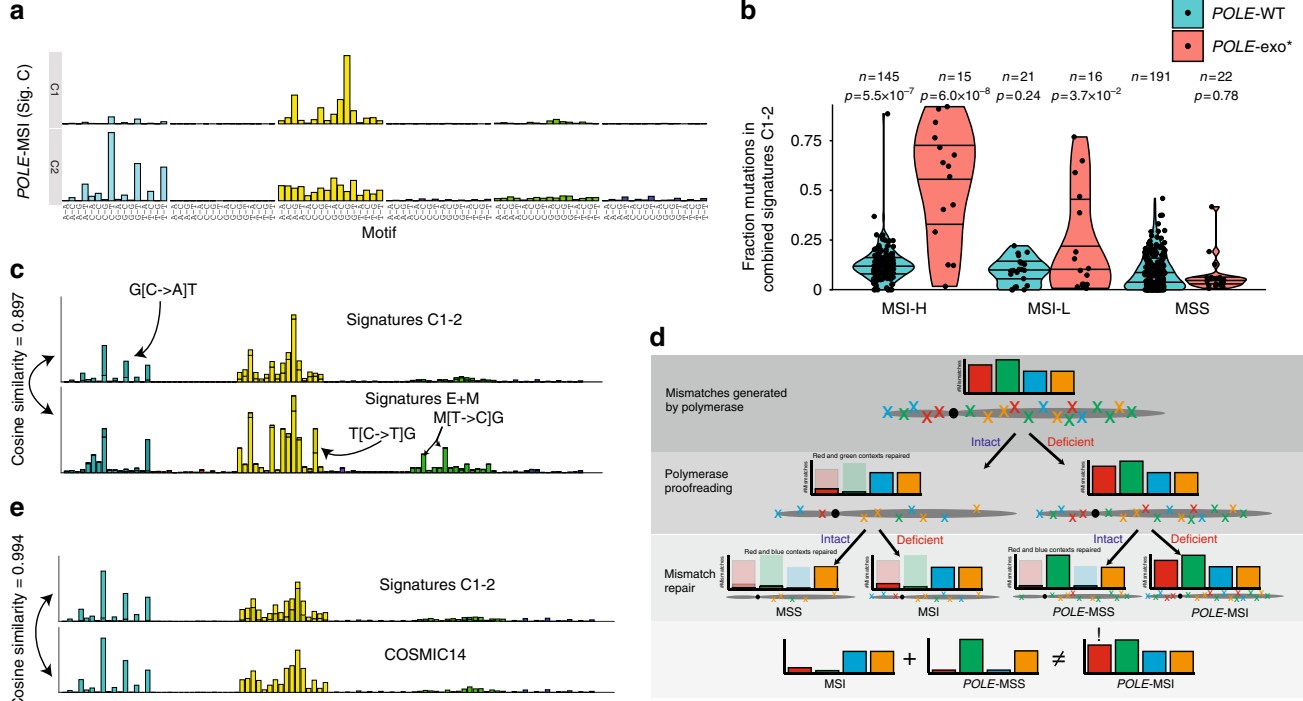

**Fig. 3** Identification of mutational signatures associated with paired loss of *POLE* proofreading and MMR. **a** SignatureAnalyzer identified two mutational signatures associated with paired *POLE*-exo mutations and MSI (C1 and C2). **b** The fraction of mutations across all cases assigned to signatures C1 and C2, segregated by *POLE*-exo* and MSI status. The distribution of each group is compared to that of polymerase wild-type MSS samples using a one-tailed rank-sum test. Horizontal lines within each group represent quartile values. **c** The five signatures of individual repair defects (signatures M1, M2, E1, E2, and E3) were combined to maximize cosine similarity to a combination of the two signatures of paired defects (C1 and C2). Each set of signatures retains several unique peaks, highlighted with arrows. M represents either A or C nucleotides. **d** Model highlighting the non-additive mutational signature contributions of combined mutational processes. **e** A linear combination of the signatures C1 and C2 closely resembles COSMIC signature 14

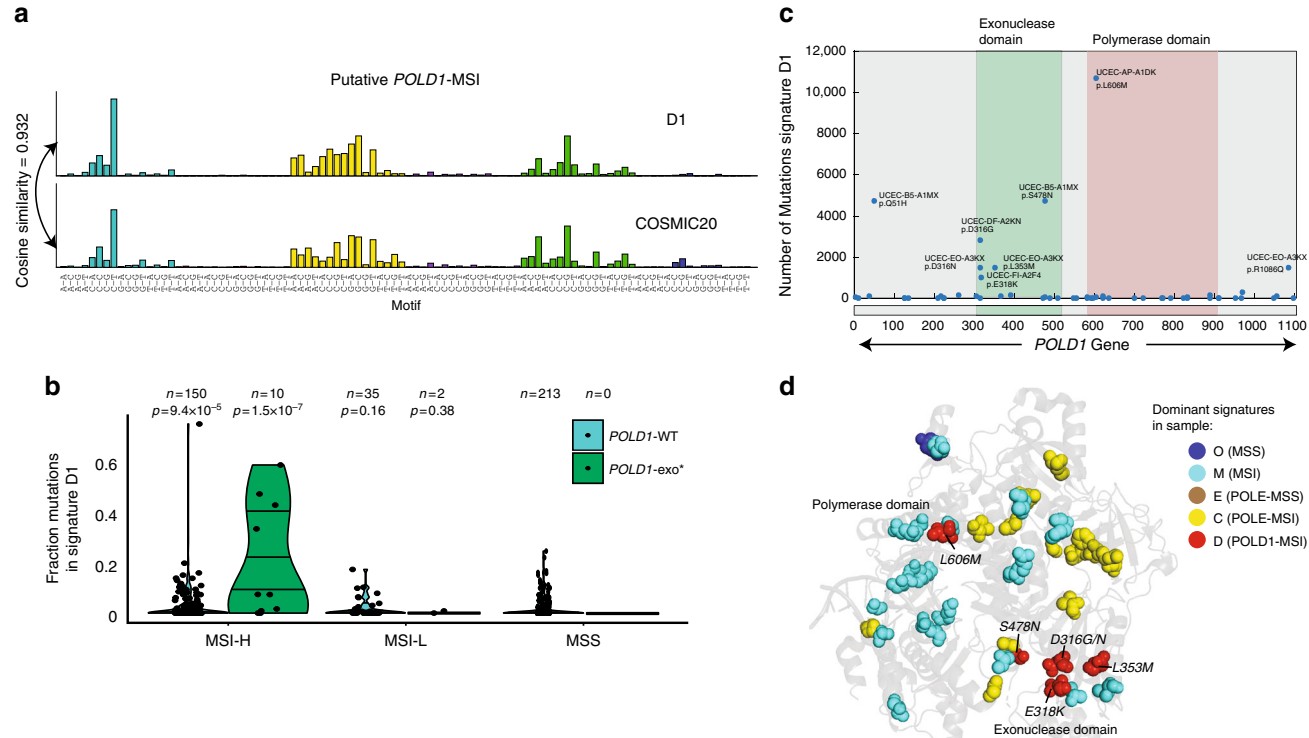

**Fig. 4** A distinct mutational signature is associated with paired loss of *POLD1* proofreading and MMR. **a** The 96 trinucleotide mutational spectrum of signature D1 and comparison to COSMIC signature 20. **b** The fraction of mutations assigned to signature D1 across the cohort, segregated by POLD1-exo* and MSI status. The distribution of each group is compared to that of polymerase wild-type MSS samples using a one-tailed rank-sum test. **c** Association of signature D1 with mutations in *POLD1*. **d** Location of *POLD1* mutations mapped to the corresponding location on the yeast *POLD1* homolog (PDB ID: 3IAY). Amino acids are color coded by the dominant mutational signature in the sample

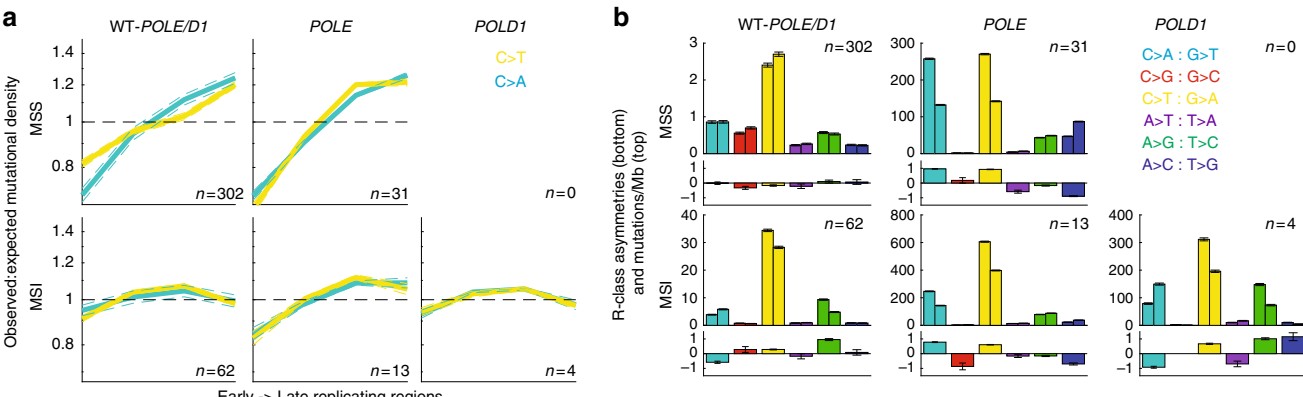

**Fig. 5** Genomic properties of repair deficiency. **a** Replication timing profiles of tumors with all combinations of polymerase and MMR deficiencies. Subsets of tumors with spectra dominated by the signature corresponding to each defect were selected (Supplementary Data 6). No *POLD1*-MSS samples were present in this data set. The *y*-axis shows enrichment in mutational density in one of four replication timing quartiles normalized by the expected density, assuming a flat background mutation rate. Only the two most frequent substitutions across all signatures, C>T and C>A, are shown. **b** Replication strand (R-class) mutational asymmetries. The top panel of each subplot shows stranded mutational densities for each base pair change. In each set of twin bars, the left bar represents the rate of mutations where the A or C of the A:T or C:G base pair is predicted to be on the leading strand template, and the right bar where the A or C is on the lagging-strand template. In the lower panel, the $\log_2$ ratio of these bars is shown, representing the asymmetry between the leading and lagging strands for this substitution type

effect of the *POLE*-exo mutation. However, the remaining 4/6 samples displayed typical MSI spectra dominated by signatures M1 and M2, suggesting their respective *POLD1* alterations (R306C, V392M, V477M, and V489A) do not impact exonuclease function. Notably all eight signatures described above can be clearly seen in the mutation spectrum of each individual patient with the corresponding defect (Supplementary Figure 9).

To extend and validate our findings, we expanded our analysis beyond endometrial cancer. First, we examined somatic mutation data from ten *POLE/POLD1*-mutated glioblastoma tumors[8] (seven with *POLE*-exo* and three with *POLD1*-exo*) from patients with bMMRD. We combined these cases with 61 TCGA endometrial cases that had either a *POLE* or *POLD1* proofreading mutation, including both MSI and MSS cases, and reanalyzed the

mutational signatures. This analysis rediscovered signatures C1, C2, and D and found that both signatures C1 and C2 were exclusive to the seven bMMRD samples with *POLE*-exo\*, and signature D to the three samples with *POLD1*-exo\* (Supplementary Figure 10, 11). Next, we expanded our search to tumors from other TCGA cohorts and identified signatures C1 and C2 in one stomach tumor (1/400), one pancreatic tumor (1/166), and a single low-grade glioma (1/516) (Supplementary Figure 12). Indeed, all three of these cases had a *POLE*-exo\* mutation and >10,000 somatic exome mutations. The stomach sample displayed *MLH1* methylation and was MSI-H. MSI status was not available for the other two samples; however, both the stomach and glioma tumors had high levels of MS indels (296 and 98, respectively), indicative of MMR deficiency[13] (Supplementary Data 5). Unfortunately, the number of MS indels could not be determined for the pancreatic tumor due to low purity. Thus, the *POLE*-MSI signature is not exclusive to endometrial tumors, although the frequency is much reduced in non-endometrial (<1% of samples) compared to endometrial cancers (19/531 samples, ~3.6%).

**Genomic properties and covariates**. To further investigate the mechanistic underpinnings of the putative *POLE/POLD1*-MSI signatures, we examined the correlation of DNA replication timing with mutational densities in tumors enriched with each signature (Methods section). Although most tumors exhibit increased mutation frequency in late-replicating regions[22,23], the mutational profiles of MMR-deficient tumors do not show such a correlation[2], suggesting that the presence of intact mismatch repair is responsible for this variability. The relationship between late replication and increased mutational density has been shown

to be particularly strong in *POLE*-mutated tumors[2]. We observed similar trends here: (i) the genomes of MSS tumors with a *POLE* proofreading mutation showed increased mutational burden in late-replicating regions (~20% increase from mean; Fig. 5a); and (ii) there was no significant correlation with replication timing in MSI tumors with WT *POLE* exonuclease domain ($p = 0.21$ and 0.42 for F-test on slope of C>A and C>T profiles; Fig. 5a). However, genomes from *POLE*-MSI tumors did show an enrichment of mutations in later-replicating regions, even in tumors with negligible contributions from signature E (Supplementary Figure 13), suggesting that MMR alone is not the sole contributor to replication-associated variation (Fig. 5a). *POLD1*-exo\* genomes did not show the same enrichment (F-test $p = 0.49$ and 0.55; Fig. 5a), perhaps indicating that *POLE*-related errors are unusually biased toward late-replicating regions.

We next characterized replicative (R-class) strand asymmetries. It was previously observed that loss of either MMR or *POLE* proofreading resulted in replication strand-biased patterns of mutations[19,24], suggesting that asymmetric patterns of mismatches introduced during leading- vs. lagging-strand synthesis are only balanced when both *POLE* and MMR mechanisms are functional. These trends were confirmed in this data set, as both *POLE*-MSI and *POLD1*-MSI tumors exhibited strong R-class strand bias (Fig. 5b). Examining the replication asymmetries in various contexts, we noted that the asymmetry profiles of *POLE*-MSI genomes most closely resembled those of *POLE*-MSS genomes, consistent with error-prone mutant *POLE* contributing the majority of mutations on the leading strand template. On the other hand, the profiles of *POLD1*-MSI genomes were more similar to those of MSI tumors (with WT *POLD1*), perhaps suggesting the mutations unveiled by faulty MMR in the presence of functional polymerase proofreading are more heavily

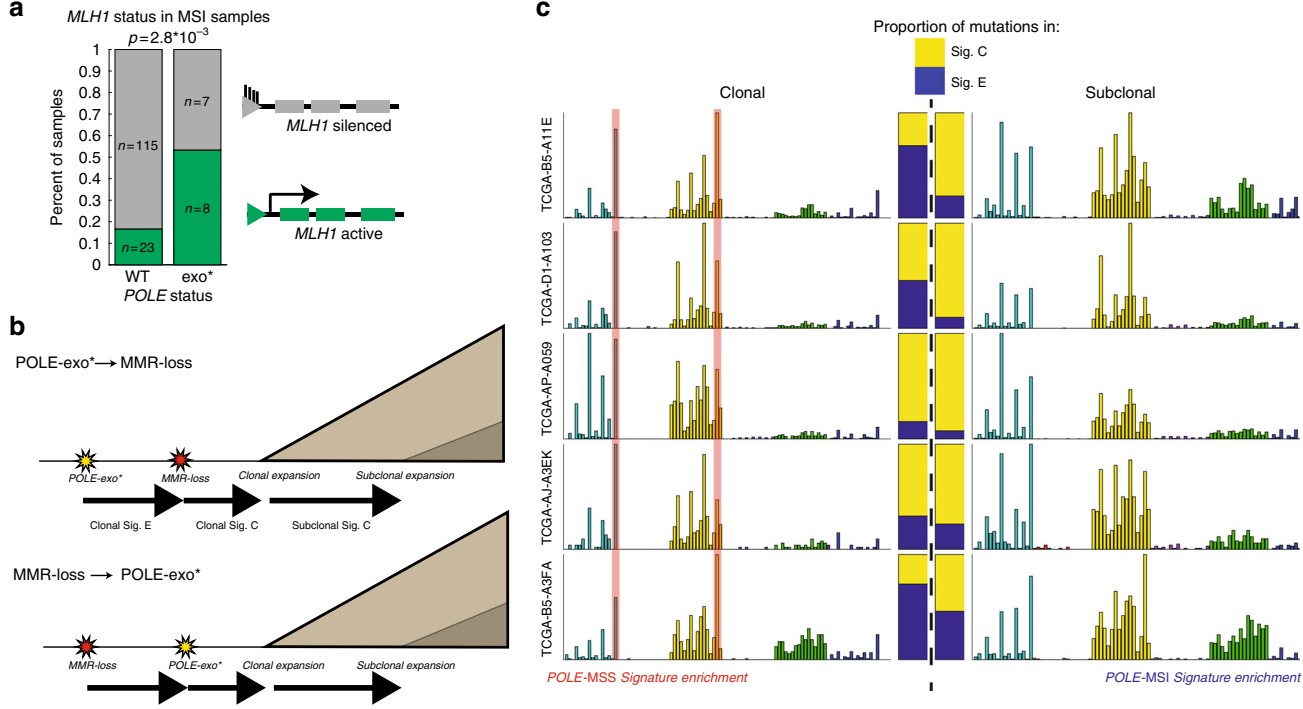

**Fig. 6** Timing of *POLE*-exo\* and MSI events. **a** Frequency of *MLH1* silencing as the mechanism of MMR deficiency in *POLE* WT vs. *POLE*-exo\* cases. 153/160 MSI-H samples with MLH1 silencing data available are shown. **b** Model representing accumulation of clonal and subclonal mutations based on the order of *POLE* and MSI events. **c** Mutational spectra of the five cases displaying significant enrichment of signature E (*POLE*-exo\*) mutations in the clonal subset of mutations, suggesting that the *POLE*-exo\* event preceded acquisition of MSI in these samples. Stacked bars represent the relative contributions of signatures E and C to the clonal and subclonal spectra of each patient. In each case, signature E (blue) is enriched in the clonal spectra

contributed by *POLD1* during lagging-strand synthesis, a hypothesis also proposed by a concurrent study[25].

**Order of repair-deficiency events**. We next sought to understand the order in which loss of MMR function and loss of *POLE/POLD1* proofreading function occurred in tumors. It is possible that loss of *POLE/POLD1* proofreading could result in a mutation in an MMR gene leading to loss of MMR pathway function, or conversely, that MSI could result in a *POLE/POLD1* exonuclease domain mutation. The majority of cases of sporadic MSI are due to *MLH1* promoter methylation,[3] and if MMR loss precedes a *POLE/POLD1*-exo mutation, then we may expect a similar proportion of *MLH1* methylation in *POLE* WT vs. *POLE*-exo* MSI samples. However, the percentage of MSI patients with *MLH1* silencing falls from greater than 80% in *POLE*-WT cases to less than 50% in cases with *POLE*-exo* (Fig. 6a, Fisher's exact $p = 2.8 \times 10^{-3}$), providing evidence against an exclusively MSI-first model.

We next considered the clonality of the mutations generated in *POLE*-MSI samples. While the alterations in the MMR or *POLE* genes in these tumors are typically clonal (Supplementary Figure 1), we reasoned that the earlier of the two events may contribute clonal mutations from its respective signature (E1−3 from a POLE-exo* event or M1−2 from an MSI event) prior to the occurrence of the second event (after which mutations from the signatures of concurrent *POLE* and MSI loss [C1 and C2] will accumulate, Fig. 6b). Similar approaches were applied to an aggregated cohort in order to infer whether signatures were active early or late in tumor evolution,[14,26] and a similar analysis of a bMMRD *POLE*-exo* case by a concurrent study found an enrichment of COSMIC signature 14 in the subclonal mutations of the tumor[9]. Therefore, we searched for enrichment of signature E (*POLE*-MSS) or M (MSI) among the clonal mutations of 19 cases with substantial contributions (at least 100 mutations and at least 35% of total) of signatures C1 and C2. Using this approach, we were able to identify five (of the 19) samples in which signatures E1−3 were significantly (Methods) overrepresented among the clonal mutations, consistent with a *POLE* mutation preceding the acquisition of MSI (Fig. 6c, Supplementary Figure 14 and Supplementary Data 7). In these cases, when mutations were separated into clonal and subclonal populations (Methods section), the characteristic signature E peaks, T(C>A)T and T(C>T)G, were overrepresented among the clonal mutations whereas subclonal mutations more closely resembled the spectrum of signatures C1−2, suggesting that the *POLE* mutation occurred prior to loss of mismatch repair. Surprisingly, we noted that 3/5 of these cases displayed *MLH1* silencing as the likely mechanism of mismatch repair deficiency, suggesting that even when *POLE*-exo* precedes acquisition of MSI, loss of MMR function is not always due to *POLE*-mediated mutagenesis of an MMR gene. Rather, dysregulation of methylation pathways or positive selection of MMR-deficient clones due to the *POLE* phenotype (e.g., if leaving *POLE*-induced mismatches unrepaired allows faster completion of S phase) may drive MSI acquisition in these cases.

In the remaining 13/19 cases, we could not identify a clear enrichment of signature E or M among the clonal mutations. In cases where the *POLE*-exo* and MSI events occur in close succession, very few signature E or M mutations would accumulate before the second event, and thus identification by this method would be poorly powered. Notably, we were not able to identify cases with significant enrichment of clonal signature M mutations (which would be expected if MSI arose first). However, the mutation rate of signature C is dramatically higher than that of signature M, and their preferred contexts are largely

overlapping; therefore, our power to detect the presence of clonal signature M mutations is limited.

## Discussion

Cells are exposed to numerous mutagenic processes across their lifetime. Replication errors are an important source of point mutations across the genome, and replication fidelity is typically enforced through dual activity of polymerase proofreading and the MMR pathway. Loss of either of these functions in isolation is associated with increased mutation burden, characteristic mutational signatures, and unique clinical properties. Here, we characterize three mutational signatures that operate in the context of concurrent loss of *POLE/POLD1* proofreading and MMR function. These signatures are closely related to COSMIC signatures 14 and 20, and our data provide a mechanistic basis for the activity of these signatures across several tumor types. Given the increasing role of tumor DNA repair deficiency as a predictive clinical biomarker—including the role of MSI and *POLE/POLD1* mutations as predictors of response to immune checkpoint blockade[27–29]—a deeper understanding of the mutational processes in these tumors may ultimately inform clinical decision-making.

Importantly, the *POLE*-MSI and *POLD1*-MSI signatures characterized here do not simply represent the aggregate sum of the two processes, but rather have unique profiles that reflect the biological interaction of *POLE/POLD1*- and MMR-mediated DNA repair. Our observations demonstrate that signatures can exhibit non-additive properties that are not described by the current paradigm for understanding and modeling mutational processes, and therefore may serve as the basis for new methodologies to discover, analyze, and interpret mutational signatures in cancer. Such improved modeling of DNA damage and repair processes will become increasingly important as more cancer mutation data accumulate.

## Methods

**Calling of somatic mutations**. In order to assemble the mutation calls for our principal cohort of endometrial samples, alignments (BAM files) of the 547 TCGA exomes were downloaded from the Genome Data Commons (GDC). Somatic point mutations were called using Mutect[30] and filtered using D-ToxoG[31], and indels were called using indelocator[32] (all available at http://www.broadinstitute.org/cancer/cga). All calls were filtered using a panel of normals. Indels in microsatellite loci were called using MSMutect[13].

**Identification of repair defects**. To identify patients with *POLE*-exo* and *POLD1*-exo*, we searched for mutations in the associated domain using mutation calls before panel-of-normal filtering. *POLE*-exo* and *POLD1*-exo* patients were identified as those with a missense mutation called within the exonuclease domain of the polymerase (codons 268–471 and 304–517, respectively[12]). All mutations falling outside this interval, mutations affecting splice sites, or leading to loss of function were discounted. Microsatellite instability status, as defined by Bethesda Protocol classification,[11] and *MLH1* silencing was determined as previously described[33] from DNA methylation array data publicly available from the GDC.

**Estimation of cancer cell fraction**. By leveraging copy number and mutation data, ABSOLUTE[34] is able to estimate the purity and ploidy of samples. Based on this information, it is able to provide the cancer cell fraction (CCF) of each mutation, as well as a confidence interval for the estimate. Sixteen samples for which these measurements could not be determined (due to low purity, low mutation counts, or lack of discrete copy number alterations) were excluded from further analysis, leaving the 531 endometrial samples that constituted our analysis set.

**Signature extraction in TCGA UCEC samples**. One of the significant challenges of de novo signature extraction in a cohort with heterogeneous mutation burdens is the increased weight of hyper- or ultra-mutant samples on the discovered signatures, which is particularly pressing in the UCEC cohort with many ultra- or hyper-mutant samples. To minimize this effect in our analysis, SignatureAnalyzer was modified in two aspects to accommodate the large mutation burden of *POLE* and MSI samples. First, we used a penta-nucleotide sequence context for single-nucleotide variants (SNVs) by considering extra contexts

at the −2 and +2 position around the mutated base, which allows 1536 channels (six possible substitutions, and 256 possible pentamer contexts centered on a given base) compared to 96 channels in the conventional trinucleotide sequence context. The use of an increased number of channels (1536 vs. 96 channels) with additional information from the pentamer context enables a better separation of mutational signatures with similar spectra when considered in 96 channels (such as signatures E1 and E3, which correspond to two different *POLE* hotspot mutations). Second, to increase discrimination power for MSI samples, which are enriched in short indels, we included insertions and deletions as additional features, along with SNVs, in the signature extraction. According to the size of inserted and deleted bases, we added eight additional channels: INS1, INS2, INS3, and INS ≥4 for insertions and DEL1, DEL2, DEL3, and DEL ≥4 for deletions. Here, INS1–3 (DEL1–3) denote insertions (deletions) with the size of one, two, and three bases, and INS ≥4 (DEL ≥4) represents insertions (deletions) with more than three bases. In summary, the input for the signature extraction is a mutation count matrix, where each column, representing a single patient, comprises (1) 1536 SNV channels corresponding to six possible substitutions (C>A, C>G, C>T, T>A, T>C, and T>G) and 256 possible pentamer contexts centered at cytosine or thymine and (2) four insertion and four deletion channels corresponding to sizes of one, two, three, and more than three nucleotides.

This matrix was then subjected to the Bayesian non-negative matrix factorization as previously described [14,15]. Out of six independent BayesNMF runs, four converged to the 12-signature solution, while two converged to the 11-signature solution. Since the only difference of the 12-signature solution from the 11-signature solution was a split of the APOBEC signature (signature O3) from the broad-spectrum signature (signature O2), we chose the 12-signature solution in downstream analysis.

**Signature extraction**. *in the combined cohort of TCGA UCEC + bMMRD samples* Mutation calls for ten bMMRD samples from Shlien et al.[8] were downloaded from the *Nature Genetics* journal website. Since the downloaded MAF had only non-synonymous SNVs, we also considered only non-synonymous SNVs in the 59 samples with signatures of *POLE*, *POLE*-MSI, and *POLD1*-MSI in UCEC cohort (Supplementary Data 6) and combined them with the mutations in the ten bMMRD samples. We again considered the 5-sequence contexts (1536 channels) as was done for the TCGA endometrial cohort, yielding a matrix of 1536-by-69 mutation counts. Out of SignatureAnalyzer's ten independent Bayesian NMF runs, nine converged to the 8-signature solution and one converged to the 9-signature solution. We have chosen the 8-signature solution for downstream analyses.

**Comparison of signatures**. In order to compare signatures, we projected them into 96-channel space by removing indel contributions and collapsing pentamer channels with a common trimer motif (e.g., the values of the 16 channels AC[C->T]GA, CC[C->T]GA,…TC[C->T]GT are all summed to calculate the corresponding 96-space channel C[C->T]G).

Signatures were then compared by calculating the cosine similarity between the 96-dimensional vectors. When comparing signatures with multiple components, we solved for coefficients that would maximize the cosine similarity of a linear combination of the components. For example, to compare signature C (made up of C1 and C2) to COSMIC signature 14 ($S_{14}$), we find values for $\alpha_1$ and $\alpha_2$ that maximize the quantity:

$$\frac{(\alpha_1 \mathbf{C1} + \alpha_2 \mathbf{C2}) \cdot \mathbf{S}_{14}}{\|\alpha_1 \mathbf{C1} + \alpha_2 \mathbf{C2}\| \|\mathbf{S}_{14}\|}$$

**Estimate of FDR when comparing signatures to COSMIC**. In order to estimate the probability of seeing a cosine similarity of 0.99 or higher to a COSMIC signature when fitting a linear combination of two signatures, as observed between signatures C1 and C2 with COSMIC signature 14, we employed two approaches. First, for each pair of our 12 signatures, we calculate the cosine similarity with the best COSMIC signature, excluding true-positive matches listed in Supplementary Data 4. For example, signature O1 is the typical CpG signature and is expected, in combination with any other signature, to match to COSMIC 1 with high similarity. Therefore, COSMIC 1 is excluded in comparison with signature O1 paired with any other signature. The distribution of similarities of all resulting matches is plotted in Supplementary Figure 7a. Second, to estimate the false positive rate due to mathematical overfitting without the need for identification of true-positive matches, we employed a permutation-based approach. For each signature, we shuffle the contributions of the 96 trinucleotide channels resulting in randomized signatures with equivalent sparsity to our own. We then calculate the cosine similarity to the best match in the COSMIC database. This process is repeated for 10,000 iterations to estimate the distribution of similarities of randomized signatures to the best COSMIC database match. This distribution is shown in Supplementary Figure 7b.

**Calculation of replication timing profiles**. For each combination of repair defects (*POLE*-MSS, *POLE*-MSI, etc.), we identified samples with overall spectra dominated by the corresponding signature based on the following criteria listed in

Supplementary Data 6. The coding genome was split into four bins using replication timing data from six lymphoblast cell lines[22]. For each bin, we then counted the number of substitutions of each type that occurred in a given sample. This was then compared to the expected mutation count, calculated for a given patient $p$ and substitution $S$ as:

$$n_{p,S}^{\exp} = \sum_{c \in S} \frac{\lambda_{c,p}}{N_c^{\text{global}}} N_c^{\text{bin}}$$

where $c$ is a channel of substitution type $S$ (e.g., for substitution C>A, channels AA[C>A]AA, AA[C>A]AC, …TT[C>A]TT). $N_c$ represents the number of opportunities for a mutation of channel $c$ across the genome (global) or in a given bin (bin), and $\lambda_{c,p}$ represents the mutation burden of channel $c$ for patient $p$, approximated by:

$$\lambda_{c,p} = \sum_{i=1}^{\text{nsig}} w_{c,i} h_{i,p}$$

where $w$ and $h$ are elements of the W and H matrix from the NMF extraction.

Error bars were estimated by sampling patients with replacement 1000 times, and calculating the standard deviation of the above metric across iterations.

**Calculation of replicative strand asymmetries**. Patients were selected as in the replication timing analysis. Then, mutational densities were calculated with respect to the leading strand template, as in our previous work[24], for all 12 possible stranded base substitutions. Asymmetries were then calculated as the log₂ ratio of complementary substitutions. Error bars were estimated by sampling mutations with replacement 1000 times and calculating the standard deviation of each measurement.

**Test for *POLE* and MMR deficiency event ordering**. To test the ordering of *POLE*-exo* and MMR deficiency events, we reasoned that after the first event, mutations of the corresponding signature (E or M) would be accumulated before the second event, after which signature C would accrue. Assuming both events were clonal, this would lead to an enrichment of signature E/M among the clonal mutations, and contributions should be diminished in the subclonal subset. On the other hand, if the events occurred more or less simultaneously, we should see no such enrichment, leading to the following null ($H_0$) and alternative ($H_a$) hypotheses:

$$H_0 : \frac{\lambda_E^c}{\lambda_C^c} \leq \frac{\lambda_E^s}{\lambda_C^s} \qquad H_a : \frac{\lambda_E^c}{\lambda_C^c} > \frac{\lambda_E^s}{\lambda_C^s}$$
$$H_0 : \frac{\lambda_M^c}{\lambda_C^c} \leq \frac{\lambda_M^s}{\lambda_C^s} \qquad H_a : \frac{\lambda_M^c}{\lambda_C^c} > \frac{\lambda_M^s}{\lambda_C^s}$$

where $\lambda_b^a$ represents the expected mutation count of signature $b$ with clonality $a$.

To calculate our test statistics, probabilities of each mutation having been generated by a given process were assigned as in ref. [14]. Then, probabilities were summed for signatures of a given set (e.g., $P(\text{sigM}) = P(\text{sigM1}) + P(\text{sigM2})$). For each patient, mutations were subset to clonal (mutations with 95% confidence cancer fraction > 0.75) and subclonal (95% confidence of CCF < 0.75). The number of clonal ($n^c$) and subclonal ($n^s$) mutations in each subset contributed by signatures E, M, and C was estimated by summing their respective probability values.

We then randomly sampled 1,000,000 clonal and subclonal mutation counts, $n^{c*}$ and $n^{s*}$, from Poisson distributions parameterized by $n^c$ and $n^s$ for each signature. $p$ values were then established by counting the fraction of iterations in which

$$\frac{n_E^{c*}}{n_C^{c*}} < \frac{n_E^{s*}}{n_C^{s*}}$$

for signature E and likewise for signature M. Finally, we applied a Benjamini–Hochberg multiple hypothesis testing correction across patients and determined significance with a threshold of $q < 0.1$.

**Code availability**. Code for SignatureAnalyzer (v1.1) that we used to perform mutational signature analysis (with default parameters) can be downloaded from www.broadinstitute.org/cancer/cga under an open-source license.

**Data availability**. DNA sequencing data and methylation data used in this study are available at the Genome Data Commons website (https://gdc.cancer.gov/).

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

## Acknowledgements

We thank Hui Shen and Peter Laird for providing epigenetic silencing calls. G.G. was partially funded by the Paul C. Zamecnick, MD, Chair in Oncology at MGH. N.J.H., Y.E. M., P.P., and G.G. were partially funded by G.G. start-up funds at MGH. J.K. and G.G. were partially funded by the NIH TCGA Genome Data Analysis Center (U24CA143845). M.S.L. was funded by start-up funds from the Massachusetts General Hospital Cancer Center. K.W.M. was funded by the NCI (1K08CA219504) and the Burroughs Wellcome Fund.

## Author contributions

N.J.H., J.K., Y.E.M., P.P., K.W.M., M.S.L., and G.G. designed the project. N.J.H., J.K., Y.E. M., and M.S.L. carried out analyses. D.R., D.L., and I.L. prepared mutation calls and ABSOLUTE results. A.K. analyzed *POLD1* protein structure. J.M.H. advised on analysis. N.J.H., J.K., Y.E.M., K.W.M., M.S.L., and G.G. wrote the manuscript. P.P., K.W.M., M.S. L., and G.G. supervised the project.

## Additional information

**Competing interests:** The authors declare no competing interests.

