## [Peer Review File · Nature Communications]

Editorial Note: This manuscript was previously reviewed at a journal not partaking in a Transparent Peer Review scheme. Please find below the Reviewer Comments & Author Rebuttals while the paper was at Nature Communications.

REVIEWERS' COMMENTS:

Reviewer #3

The rewriting/trimming of results has made this a better manuscript than the original submission. It's still a bit of a niche advance, however. But one of interest to the mutation signatures field.

2 quibbles:

[1]

ref 3 point 5: this still reads like a circular argument:

the authors write that what they have is "Situation 2" not "Situation 1" based on some samples being composed almost entirely from Signature C1+C2 (putatively MMR X POLE interaction). But this is assuming that C1+C2 represent only the MMR X POLE interaction (without admixture of 'normal' POLE). And the argument for this is because they have "Situation 2". Without more confident timing of the relative events of MMR failure vs. the individual mutations its v difficult to make this argument.

[2] minor:

"when summed, Signatures C1 and C2 closely resemble COSMIC Signature 14" ... this is misleading because it's not just a sum - they use regression to fit the coefficients of C1 and C2, so it's not as straightforward as this sentence implies.

REVIEWERS' COMMENTS:

Reviewer #3

The rewriting/trimming of results has made this a better manuscript than the original submission. It's still a bit of a niche advance, however. But one of interest to the mutation signatures field.

2 quibbles:

[1]

ref 3 point 5: this still reads like a circular argument:

the authors write that what they have is "Situation 2" not "Situation 1" based on some samples being composed almost entirely from Signature C1+C2 (putatively MMR X POLE interaction). But this is assuming that C1+C2 represent only the MMR X POLE interaction (without admixture of 'normal' POLE). And the argument for this is because they have "Situation 2". Without more confident timing of the relative events of MMR failure vs. the individual mutations its v difficult to make this argument.

We agree with the reviewer on the need to clarify this argument to avoid the appearance of circularity. Our logic can be divided into two sequential claims:

Claim 1) There exist patients with spectra dominated by POLE-MSI mutations (i.e. nearly all mutations were acquired after acquisition of the second event).

Claim 2) Assuming there exist patients with a pure spectra generated by joint POLE-MSI, we will extract a signature corresponding to this mode of mutagenesis.

The concern of the reviewer is that to prove (1) we must already know that our Signatures C1 and C2 are pure representations of POLE-MSI, the very thing we are trying to demonstrate in (2). However, the evidence for (1) is not contingent on our ability to extract a pure signature of POLE-MSI. Assume (by contradiction) that all patients accumulate substantial numbers of mutations before acquisition of the second deficiency. These mutations will follow the same spectrum as in patients that never acquire a second event (i.e. Signature E or M, depending on the first deficiency). We can then consider two cases:

a) The second event is clonally acquired. In this case clonal mutations will have some substantial component of Signature E or M in addition to the pattern observed in the subclonal mutations (which are generated after both deficiencies are acquired).

b) The second event is subclonally acquired. In this case, clonal mutations should be purely Signature E or M, and subclonal mutations will be a mix of Signature E or M and the true POLE-MSI pattern.

However, the counter examples given in the previous rebuttal (figure below) contradict both these cases. They instead show no enrichment of Signatures E or M in the clonal mutations compared to the subclonal mutation pattern, corresponding to a situation in which few mutations are accumulated before the second hit. Note that this argument revolves around purity of Signatures E and M (of which we can be assured by the many endometrial samples with only a single repair deficiency) but needs not make assumptions about Signature C. In addition, we consider only mutations in which we are highly confident of their CCF (as described in our Methods), and other mutations are discarded for this analysis.

Therefore, claim 1 is supported by the data without needing to assume claim 2 and hence there is no circularity in our rationale for claim 2 (which we described in the previous response).

[2] minor:

"when summed, Signatures C1 and C2 closely resemble COSMIC Signature 14" ... this is misleading because it's not just a sum - they use regression to fit the coefficients of C1 and C2, so it's not as straightforward as this sentence implies.

We thank the reviewer for this point. We have altered the sentence to read:

Comparing the C1 and C2 signatures to the COSMIC signature database, we found that, when summed using coefficients fitted to maximize the cosine similarity, Signatures C1 and C2 closely resemble a signature of unknown etiology, COSMIC Signature 14 (Figure 3e), more than expected by chance (Supplementary Figure 7a,b; Supplementary Data 4).